# Analysis of Plasma Ion Distribution and Dust Collection Efficiency of Carbon-Brush Air Purifiers

Yong Sun Kim [1], Hong Gun Kim [1], Sang Cheol Ko [2] and Lee Ku Kwac [2,*]

[1] Institute of Carbon Technology, Jeonju University, Jeonju 55069, Republic of Korea
[2] Graduate School of Carbon Convergence Engineering, Jeonju University, Jeonju 55069, Republic of Korea
* Correspondence: kwac29@jj.ac.kr; Tel.: +82-63-220-3063

**Abstract:** In recent years, many studies on air purifiers have been conducted, as particulate matter and virus issues have emerged. In this study, the ion concentration distribution in an air purifier that applies a high voltage was investigated through simulation. For a single carbon brush that applied a high voltage of –8.5 kV, the simulation results of the ion concentration distribution in the ground direction were compared with the result of the experiment and an error of 4.3% was observed, thereby confirming the reliability of the simulation. On this basis, the ion concentration distribution was calculated according to the number and location of the brushes. In addition, the charging number was calculated by applying the charging mechanism to the distributed dust particles, and the dust collection efficiency was calculated by conducting particle multiphysics analysis. The dust collection efficiency increased from 0.5% to 1% as the number of brushes increased, and the dust collection efficiency was 82% when there were two brushes and 83% when there were four brushes. In the proposed modeling, the location of the brushes is more important than the number of brushes. These results are expected to provide more accurate design information for the number and location of brushes applicable to an air purifier.

**Keywords:** computational fluid dynamics (CFD); carbon-brush; collection efficiency; air-purifiers; ion concentration





## 1. Introduction

Recently, various air purifiers have been manufactured and distributed in response to increasing interest in air purifiers due to concerns over airborne particulate pollutants and viruses. Air purifiers can be divided into physical filter types, such as high-efficiency particulate air (HEPA) filters, and electrical filter types that use static electricity or ion emission [1–8].

For physical filters, various types of filter media are available, and studies have been conducted to improve dust collection performance through flow analysis and experiments. In the case of air purifiers that emit ions, however, experimental studies have focused on analyzing filter performance because of the difficulty in analyzing the ion distribution [9–11].

In the case of air purifiers that emit ions by applying a high voltage, the wire type and metal needle type have mainly been used. For ion-emitting air purifiers, the application of a high voltage to a wire or a metal needle causes the dielectric breakdown of air, and the current formed between the discharge electrode and ground electrode by ionization reaction generates ions, thereby charging dust particles. The charged dust particles change to an anode or a cathode due to the movement of free electrons, and they are collected through the attractive or repulsive force generated by the electric field of the dust-collecting component. In this instance, the metal needle to which high voltage is applied generates a large amount of ozone. Because this type of air purifier generates ozone in large quantities, an ozone removal module is required in addition to the air purifier module [12–14].

Jeong et al. conducted a comparative analysis of the number of ions generated according to the application of a high voltage, and experimentally analyzed the ion generation

optimization conditions in a corona discharge static eliminator. As a result, the equation was derived under the AC application conditions, and the characteristics of the voltage difference between AC and DC were identified [15,16]. Kim et al. conducted an experimental study on improving dust collection efficiency by analyzing the efficiency of an activated carbon filter according to the increase in ions and the dust collection efficiency according to the intensity of the electric field. As a result, the antibacterial performance and dust collection efficiency of the ionizer were secured, and the possibility of a high-efficiency multifunctional air purifier was experimentally secured [17,18]. Yoa conducted an experimental study on improving the dust collection efficiency of an ionizer by analyzing the pressure loss according to the flow velocity and collector. As a result, it was possible to increase the efficiency of fine dust particles by introducing the dielectric and ionizer method, and to obtain the possibility of large-capacity gas processing and high-efficiency device miniaturization with low pressure loss at a high flow rate [19,20]. Yoon measured the ozone concentration according to the applied voltage in an air purifier that applied carbon brushes, and experimentally analyzed the dust collection efficiency according to the number of brushes. As a result, the correlation between the number of carbon brushes, voltage, and ozone was derived, and the indoor air purifier standards of the Korea Air Purifier Association were satisfied [14]. Milad et al. conducted a study on a PM collector to which high-capacity ESP was applied for toxicity research. At a flow rate of 75 lpm and an applied voltage of +12 kV, they achieved particle collection efficiency of more than 80% for almost all particles with a size range of 0.015–2.5 μm, while the ozone concentration was 17 ppb [21]. Taghvaee et al. developed a new method to generate particulate matter in an inhalation exposure study. In order to investigate particulate matter properties, PM sources that are physically and chemically stable over a wide range of processes were generated [22]. In addition, recent studies have been conducted studies on the spread of harmful particles such as COVID-19 and the effects of pollutant exposure [23,24]. Most of these studies conducted experimental studies on the generation method and collection efficiency of ions in the purifier.

This study aims to secure the reliability of the simulation by conducting simulations and experiments at the same time for air purifier research carried out as an experiment. In the present study, the number of ions generated in an air purifier with carbon brushes was analyzed using simulation. To confirm the reliability of the simulation, a constant DC with high voltage was applied to a brush and anions were measured according to the distance from the brush using an ion counter, and the results were compared with the ion generation and distribution results by simulation.

In addition, the dust collection efficiency was calculated using the secure ion distribution and electric field. The particles applied for the dust collection efficiency were the 0.3 μm standard test particles used by the Korea Air Cleaning Association, and the charging number was obtained by calculating the charging mechanism. Computational fluid dynamics (CFD) analysis was used to analyze the particle behavior, and a typical air purifier flow velocity of 0.9 m/s was applied in the spatial field. Particles flowed along the streamline simulated in CFD. Conditions were given so that the particles could be collected on the ground using the repulsive force of the dust-collecting component according to the potential difference of the electric field analysis, and the dust collection efficiency was compared. Finally, the collection efficiency was compared to confirm the design information regarding the number and location of brushes. Through this study, it will be possible to evaluate the optimal location and number of brushes.

## 2. Mathematical Basis for Simulation

The simulation was performed as shown in Figure 1. To calculate the dust collection efficiency of an air purifier that applies a high voltage, it is necessary to calculate the charging number for dust. Several variables are required to calculate the charging number. Table 1 shows the independent variables determined under indoor conditions. Dependent

variables to be calculated from simulation included the ion concentration ($N_i$) and the electric field (E).

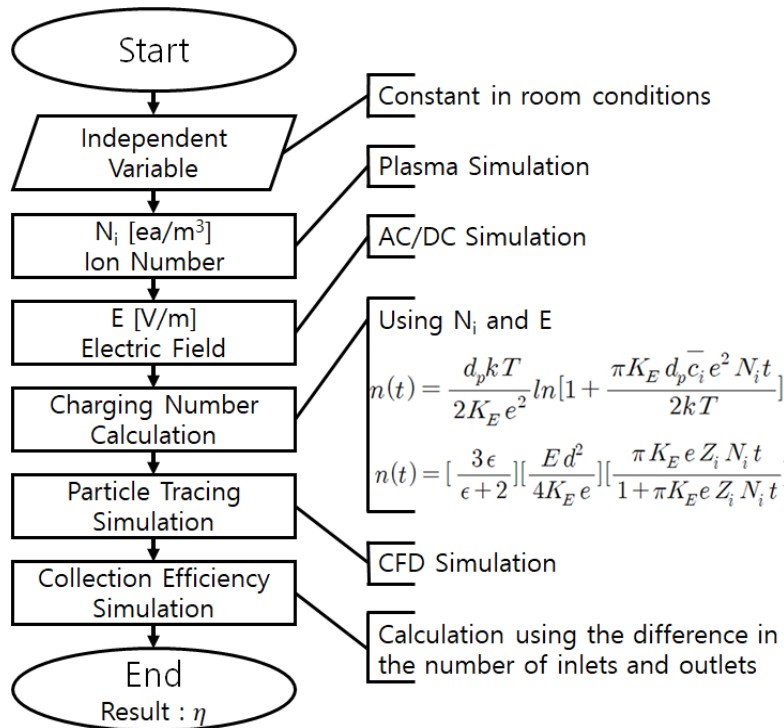

**Figure 1.** Simulation flow chart.

Plasma analysis was conducted to determine the ion concentration distribution ($N_i$), and AC/DC analysis was conducted to obtain the electric field (E). The charging number for particles was calculated using the dependent variables ($N_i$ and $E$). Finally, particle tracing analysis was conducted using the flow field and the streamline obtained for dust collection efficiency evaluation. The number of dust particles filtered by the flow field and electric field was identified, and the dust collection efficiency was calculated. The dust collection efficiency of the air purifier was compared with the experimental values under different conditions.

For dust particles to be sufficiently charged, a high ion concentration is required in the electric field. When a voltage higher than the firing voltage is applied between the brush and the ground, electrons are separated from gas molecules in the air by the strong electric field and dust particles are charged as the gas becomes cations and anions are discharged. When the energy of the electrons is lower than the ionization potential, it can excite atoms or gas molecules. For atoms or molecules that were already in the excited state before the collision, ionization can even be caused by a collision with electrons with energy levels lower than the ionization energy through the mechanism shown in Equation (1) or (2). $A^*$ represents a neutral gas molecule and $K_e$ is the kinetic energy of an electron [25,26].

$$A^* + e^- + K_e \rightarrow A^+ + 2e^- \tag{1}$$

$$A^* + A^* \rightarrow A^+ + A + e^- \tag{2}$$

Determining the number of electric charges on dust requires the diffusion charging relation given by Equation (3) and the field charging relation given by Equation (4). Equation (3) shows the number of electric charges ($n(t)$) acquired by a particle with a diameter of $d_p$ by diffusion charging during time $t$ [20,27,28]:

$$n_1(t) = \frac{d_p kT}{2K_E e^2} \ln\left[1 + \frac{\pi K_E d_p \overline{c_i} e^2 N_i t}{2kT}\right] \tag{3}$$

where $\overline{c_i}$ is the average thermal velocity of the ions ($\overline{c_i} = 240 m/s$ at standard conditions), $N_i$ is the ion concentration, $k$ is the Boltzmann constant, $T$ is the atmospheric temperature, $K_E$ is the electrical proportional constant, and $e$ is the electron charge.

**Table 1.** Charging number equation independent variables.

| Name | Expression | Unit | Description | Reference |
|:---:|:---:|:---:|:---:|:---:|
| $k$ | $1.38 \times 10^{-23}$ | J/K | Boltzmann constant | [28,29] |
| $K_E$ | $9 \times 10^{-9}$. | $N * m^2/C^2$ | Electrical proportional constant | [28,29] |
| $\overline{c}$ | 240 | m/s | Thermal speed of ion | [28] |
| $E$ | $1.6 \times 10^{-19}$. | C | Electron charge | [28,29] |
| $\varepsilon$ | 4.81 | - | Relative Permittivity of KCl | [28,30] |
| $Z_i$ | 0.00015 | $m^2/V * s$ | Mobility of Ions | [28] |
| $u$ | 0.9 | m/s | Velocity | measurement |
| $h$ | 380 | mm | Height of the charged space | measurement |
| $t$ | 0.42 | s | Charging time | (h/u) |
| $d_p$ | 0.3 | um | Particle diameter | measurement |
| $T$ | 300 | K | Temperature | measurement |

Equation (4) is an equation for calculating field charging. When diffusion charging is negligible, it represents the number of electric charges ($n(t)$) acquired by a particle during time $t$ in the electric field $E$ with an ion concentration of $N_i$. $Z_i$ is the mobility of ions and $\varepsilon$ is the relative permittivity [27,28].

$$n_2(t) = \left[\frac{3\varepsilon}{\epsilon + 2}\right]\left[\frac{Ed^2}{4K_E e}\right]\left[\frac{\pi K_E e Z_i N_i t}{1 + \pi K_E e Z_i N_i t}\right] \tag{4}$$

Obtaining the total number of electric charges on one dust particle ($n_1(t) + n_2(t)$) requires the independent variables shown in Table 1. The time $t$ required to charge the particle can be obtained as the ratio of the height $h$ of the spatial field to the inlet flow velocity $v$. An attempt was made to determine the variables that can be calculated ($N_i$ and $E$) through simulation.

For the plasma analysis used to obtain the ion concentration ($N_i$), it is difficult to interpret ions as particles because of the limited analysis capacity and speed of the computer because the density of charged particles is $10^8$ cm$^{-3}$. Therefore, the behavior of electrons and ions was assumed to be a continuum, i.e., a flow.

By using this continuum assumption, the governing equation for gas discharge can be expressed considering the diffusion of electrons, as shown in the following equation [31,32]:

$$\frac{\partial n_e}{\partial t} = -\frac{\partial n_e v_e}{\partial x} + \frac{\partial^2 (D_e n_e)}{\partial x^2} + S \tag{5}$$

where $n_e$ is the electron density, $v_e$ is the electron movement speed, $D_e$ is the electron diffusion coefficient, and $S$ is the density of the electrons newly generated per hour by photoionization at position $x$. The electrons or ions calculated by this process act as space charges in the space between the electrode and the ground. The average velocity of the charged particles is derived from the Boltzmann equation. If the inertia of ions and electrons is neglected, it is expressed as [33,34]

$$\frac{\partial}{\partial t}(n_e) + \nabla \cdot [-n_e(\mu_e \cdot E) - D_e \cdot \nabla n_e] = R_e \tag{6}$$

$$R_e = \sum_{j=1}^{M} x_j k_j N_i n_e \tag{7}$$

where $x_j$ is the mole fraction of the species and $k_j$ is the scale factor. From this basic theory, the ion concentration ($N_i$) can be calculated by performing simulation using the Plasma module of COMSOL.

In the field charging equation, the electric field ($E$) value is the basic equation of Coulomb's law, and the electric force ($F_E$) can be obtained from Equation (8). Here, $q$ is the quantity of electric charge, $q'$ is the quantity of electric charge of the charged dust particles, $r$ is the charged radius, and $\epsilon_0$ is the permittivity [20,28].

$$F_E = -\frac{qq'}{4\pi\epsilon_0 r^2} \tag{8}$$

When the intensity of the electric field acting on dust particles is $E$, the electric force $F_E$ can be expressed as Equation (9). For simple geometry, the intensity of the electric field ($E$) can be expressed as Equation (10) according to Coulomb's law. Using these equations, the AC/DC module was used to calculate the $E$ value [19].

$$F_E = qE \tag{9}$$

$$E = \frac{F_E}{q'} = \frac{q}{r^2} \tag{10}$$

The charging number for dust was calculated using $N_i$ and $E$, which were obtained through simulation, and the particle trajectory was traced using the Particle Tracing module. In CFD analysis, the Navier–Stokes equation was used as a basic governing equation and a streamline was used in the result. Particles flow along the streamline. Charged particles are collected on the ground because of the attractive or repulsive force caused by the potential of the collector while flowing along the streamline. The motion of a charged particle can be explained as Lagrangian for an electric charge in an electromagnetic field.

$$L = -d_{mp}c^2\sqrt{1 - v{\cdot}vc^2} + ZeA{\cdot}v - ZeV \tag{11}$$

where $d_{mp}$ is the mass of the particle, $c$ is the velocity of light, $v$ is the particle velocity, $Z$ is the particle charge, $A$ is the magnetic vector potential, and $V$ is the electric scalar potential.

In an electric field, the equation of motion for a charged particle is

$$\frac{d}{dt}(m_p v) = -ZeEE + Ze(v \times B) \tag{12}$$

where $B$ is the magnetic flux density, which can be expressed as $B = \nabla \times A$ [35,36].

The particle collection results are obtained through this particle tracing simulation. On this basis, the collection efficiency can be calculated. Equation (13) shows the collection efficiency calculation method. Here, $\eta$ is the collection efficiency (%), $C_1$ is the number of particles in the flow before passing through the filter (ea/ml), and $C_2$ is the number of particles in the flow after passing through the filter (ea./ml).

$$\eta = \left(1 - \frac{C_2}{C_1}\right) \times 100 \tag{13}$$

## 3. Ion Concentration Simulation

### 3.1. Ion Concentration Distribution According to the Ground Distance

Based on the schematic diagram for the experiments, shown in Figure 2, a two-dimensional space with a size of $320 \times 300$ m$^2$ was created as an analysis space, and the number of ions was measured using the distance between the brush and the ground as a variable. In addition, plasma analysis was conducted to identify the ion concentration distribution. Figure 2 and Table 2 show the boundary conditions.

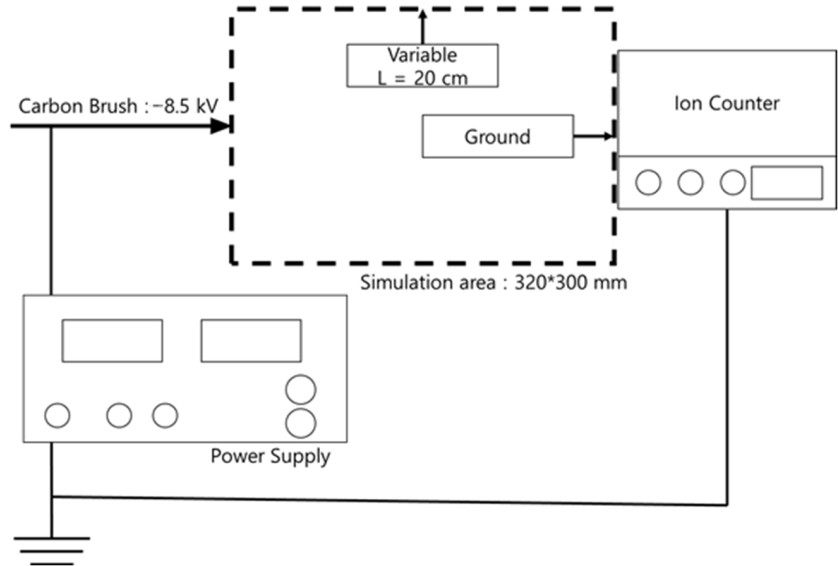

**Figure 2.** Schematic diagrams for experiments and simulations.

**Table 2.** Boundary conditions for ion concentration.

| Brush Voltage | Temperature | Absolute Pressure |
|:---:|:---:|:---:|
| −8.5 [kV] | 300 [K] | 1 [atm] |

In this setup, anions are generated by the negative voltage applied to the brush, and dust is also negatively charged. Table 3 and Figure 3 show the simulation results. The distance from the brush to the ground was defined as L, and five cases from 10 to 30 cm at 5 cm intervals were analyzed. The anions diffused in the ground direction as shown in Figure 3, and the ion concentration decreased according to the distance as shown in Table 3.

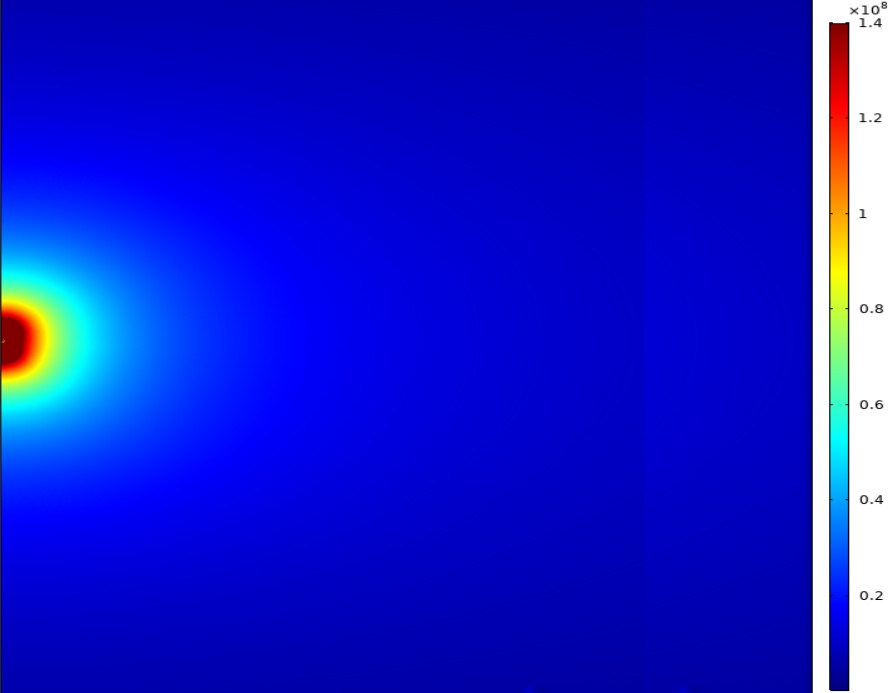

**Figure 3.** Ion concentration contour distribution.

**Table 3.** Ion concentration according to brush and ground distance.

| L [cm] | Number of Ions [ea·10$^6$/cc] |
| --- | --- |
| 10 | 1150 |
| 15 | 888 |
| 20 | 650 |
| 25 | 418 |
| 30 | 339 |

To experimentally measure the number of anions generated by the carbon brush applied to the discharge component, measurement was performed under the conditions shown in Figure 2 in a chamber with no flow. DOWAKSA's A-42 carbon fiber 24K was used as the carbon brush applied to the discharge component.

The number of anions generated from the discharge component of the carbon brush was measured three times at the same points in a straight line from the discharge component to the ground as in the simulation, and the average values were compared with the simulation results. Figure 4 and Table 4 shows the results. As in the simulation results, the ion concentration decreased as the distance increased. When the experiment and simulation results were compared, an average error of approximately 4.2% was observed.

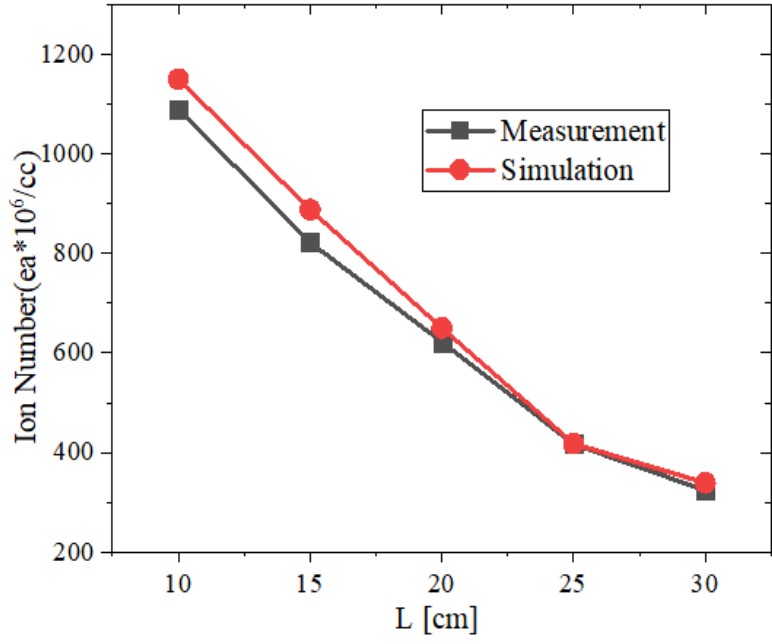

**Figure 4.** Ion number comparison between experiment and simulation.

**Table 4.** Comparison of experiments and simulations at ion concentrations.

| L [cm] | Number of Ions [ea·10$^6$/cc] | | | | |
| --- | --- | --- | --- | --- | --- |
| | 1st | 2nd | 3rd | Average | Simulation |
| 10 | 1.091 | 1.088 | 1.090 | 1.090 | 1.150 |
| 15 | 0.825 | 0.818 | 0.823 | 0.822 | 0.888 |
| 20 | 0.623 | 0.620 | 0.622 | 0.622 | 0.650 |
| 25 | 0.430 | 0.415 | 0.417 | 0.417 | 0.418 |
| 30 | 0.330 | 0.317 | 0.326 | 0.324 | 0.339 |

### 3.2. Ion Concentration According to the Ground Position

The reliability of the simulation was confirmed by comparing the simulation and experiment results. On this basis, the ion distribution according to the ground position was examined. The ion concentration ($N_i$) varied depending on the ground position. To analyze

this phenomenon, two cases were modeled as shown in Figure 5, and the number of ions generated was compared. The number of ions generated reached its peak at a position 2 mm away from the brush. Table 5 shows the simulation results.

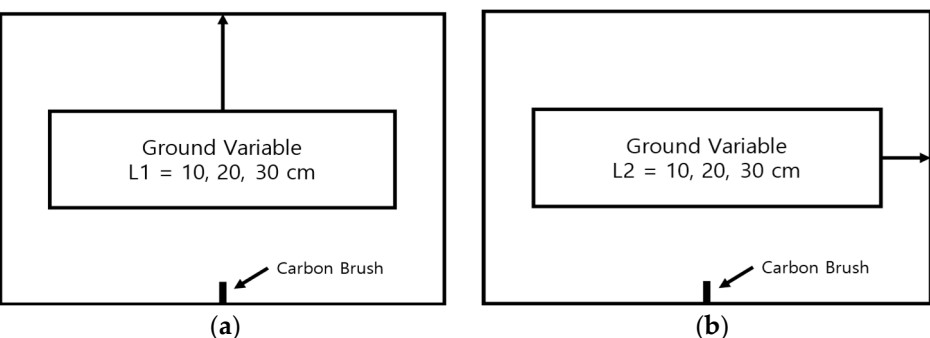

**Figure 5.** Geometry according to ground distance; (**a**) Case 1; (**b**) Case 2.

**Table 5.** Ion concentration results according to the ground position.

| L [cm] | Maximum Number of Ions [ea·$10^6$/cc] | |
| --- | --- | --- |
| | Case 1 | Case 2 |
| 10 | 4.024 | 3.872 |
| 20 | 2.481 | 2.617 |
| 30 | 1.225 | 1.119 |

The ion concentration decreased as the distance from the brush in the ground direction increased. This result appears to be due to the change in the quantity of electric charge in the space caused by the potential difference between the electrode and the ground. This result confirms that the ground position is a highly important variable for ion concentration analysis.

## 4. Air Purifier Dust Collection Efficiency Simulation

### 4.1. Ion Distribution According to the Location and Number of Brushes

To apply carbon brushes to the air purifier as shown in Figure 6, three brush configuration cases were defined for the back of the air purifier, and the ion concentration distribution according to the number of brushes was simulated. The location of the brushes was 100 mm from the bottom, and a voltage of $-8.5$ kV was applied. The flow velocity was set to 1.0 m/s, and the ground and an applied voltage of $-5$ kV were alternately designed for the dust collecting plate in the collector. The length of the air purifier module was set to 400 mm, and two to four brushes were applied depending on the case. The ground was set to the bottom where the dust collecting plate was placed so that ions could be generated in the ground direction. Based on the simulation performed, the charging number for each particle was obtained by applying Equations (3) and (4), which are the charging equations.

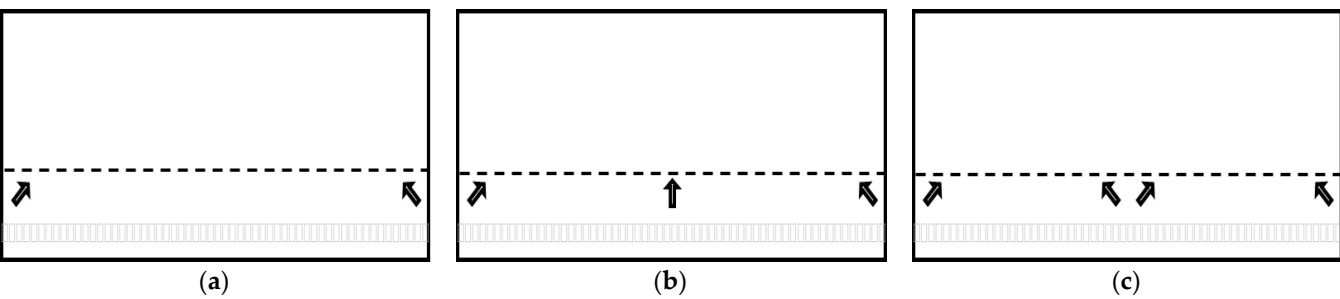

**Figure 6.** Schematic of the location of the number of brushes; (**a**) Case 1; (**b**) Case 2; (**c**) Case 3.

The ion concentration was calculated using the 100 mm line with the highest ion concentration close to the brush location as the dotted line shown in Figure 6. From the charging equations, the charging number for each case was obtained.

In Case 1, the brushes were installed in the 45° direction to face each other, and ion distribution simulation was performed. Figure 7 shows the ion concentration contours. It can be seen that the ion concentration distribution developed toward the ground direction.

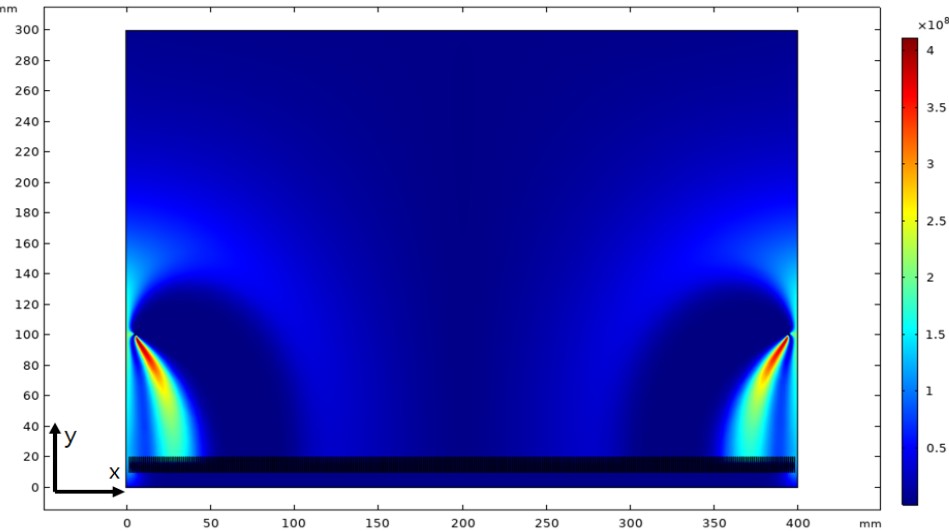

**Figure 7.** Ion concentration distribution for the brushes in Case 1.

The charging number is shown in Figure 8 according to the charging equations by selecting the line with the highest ion concentration. It can be seen that ions were generated from the brush and developed in the ground direction. In addition, the charging number showed a tendency similar to that of the ion concentration.

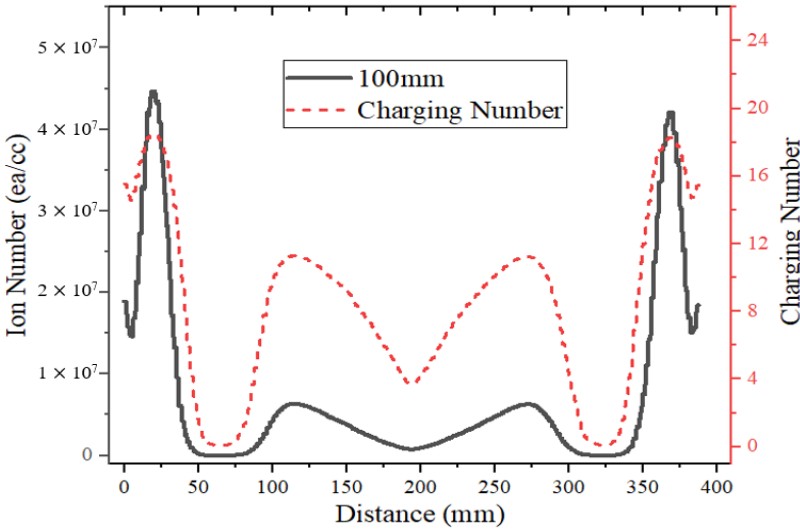

**Figure 8.** Ion concentration and charging number for Case 1.

In Case 2, a vertical brush was additionally installed in the center compared to Case 1, and ion distribution simulation was performed. Figure 9 shows the ion concentration contours. It can be seen that the ion concentration distribution developed toward the ground direction as in Case 1, and that the ion concentration developed from the brush in the center to both sides. This result indicates that ions are generated at the sharpest part when they are generated from a brush or a metal needle. In the case of brushes, ions are

generated at both ends of the fiber bundle, and thus the ion concentration distribution occurs as shown in Figure 9 [37–39].

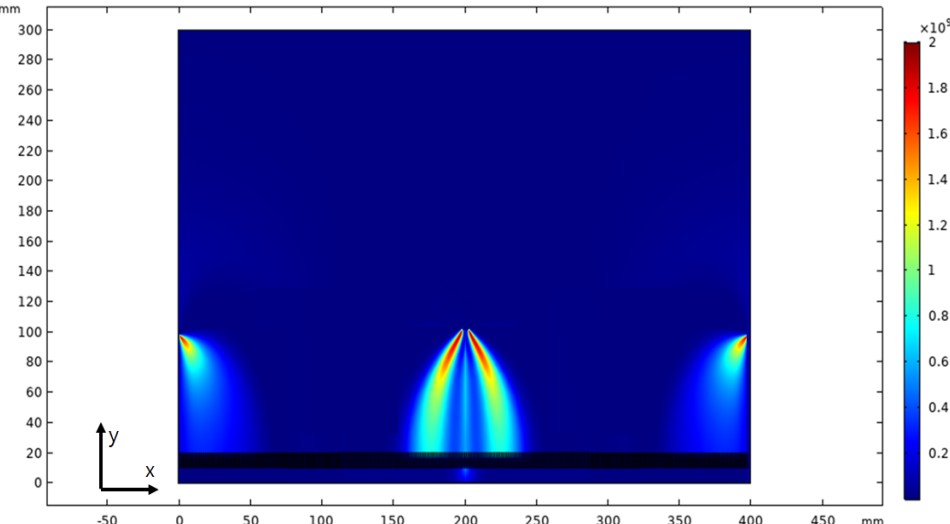

**Figure 9.** Ion concentration distribution for the brushes in Case 2.

The charging number is shown in Figure 10 on the basis of the charging equations by selecting the 100 mm line. It can be seen that two peaks occurred as the concentration of ions generated at both ends of the brush was measured. The same number of ions as in Case 1 was generated in the left and right brushes, but a high ion concentration was observed from the brush in the center because of the overlapping ion flows.

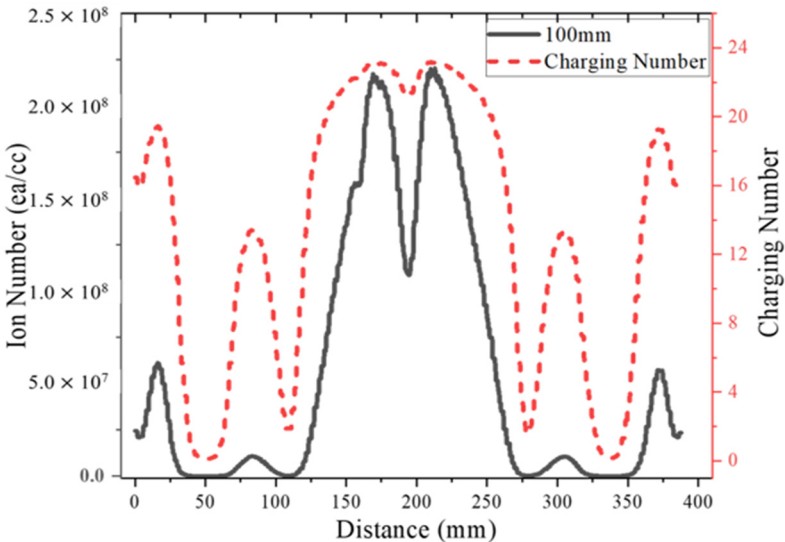

**Figure 10.** Ion concentration and charging number for Case 2.

In Case 3, two brushes in the 45° direction were additionally installed in the center compared to Case 1, and ion distribution simulation was performed. Figure 11 shows the ion concentration contours. It can be seen that the ion concentration distribution developed toward the ground direction as in Case 1, and the ions generated from the brushes in the center were dispersed because of the same polarity. The charging number is shown in Figure 12 according to the charging equations by selecting the 100 mm line.

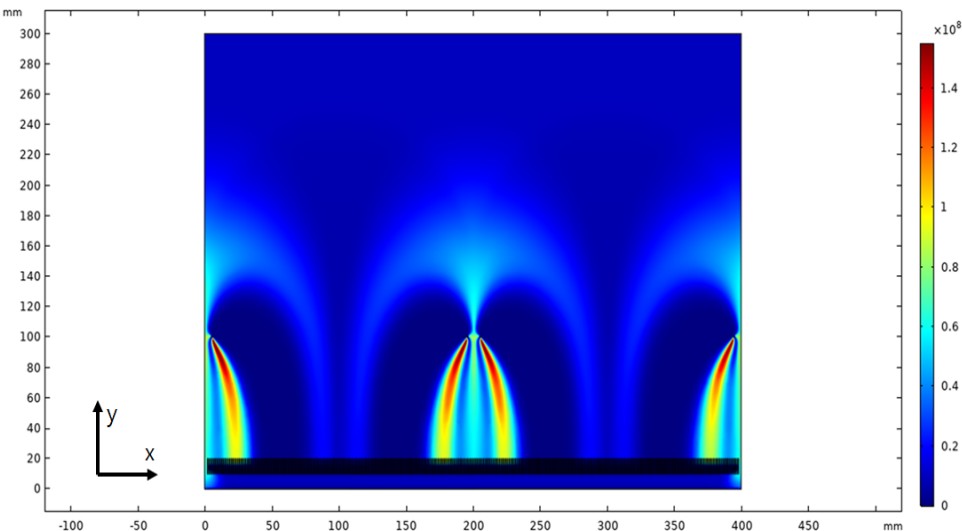

**Figure 11.** Ion concentration distribution for the brushes in Case 3.

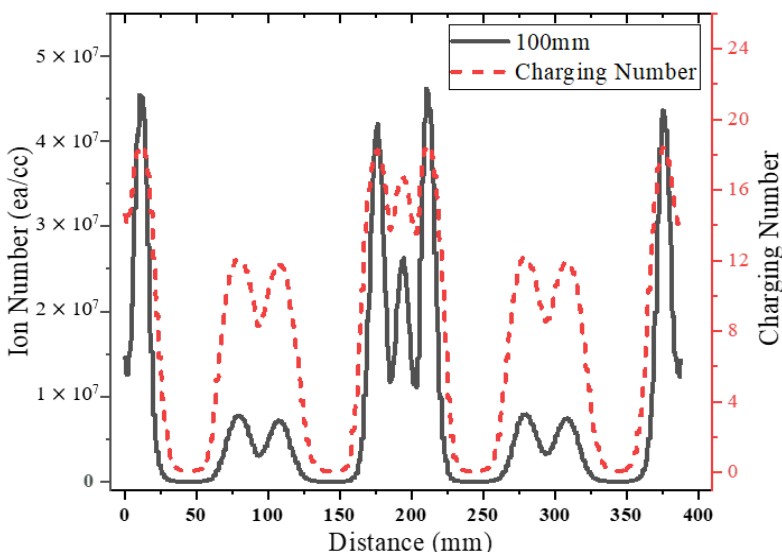

**Figure 12.** Ion concentration and charging number for Case 3.

### 4.2. Dust Collection Efficiency Simulation

To calculate the dust collection efficiency, the calculation area of the air purifier was set as shown in Figure 13, and the inside of the air purifier was simulated through multiphysics analysis for each case. The air purifier was modeled considering its horizontal symmetry, and the boundary conditions of each simulation were set as shown in Table 6. On the basis of the flow shown in Figure 1, ion concentration analysis, electric field analysis, flow analysis, and multiphysics analysis of particle behavior were conducted. In the case of the collector, the voltage and ground were alternately designed, and simulation was performed so that the particles charged by anions could be collected at the ground by means of repulsive force.

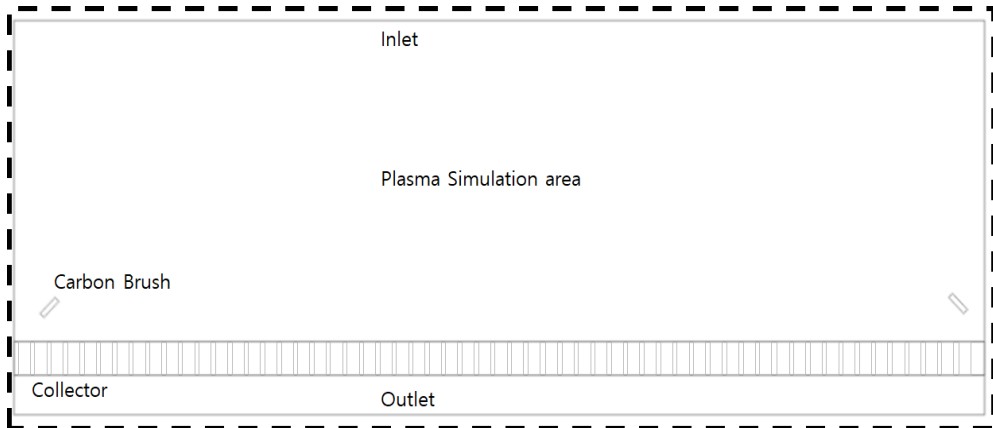

**Figure 13.** Air purifier collection simulation geometry.

**Table 6.** Boundary conditions for multiphysics analysis.

| Inlet Velocity | Brush Voltage | Collector Voltage | Inlet Particle |
|---|---|---|---|
| 0.9 [m/s] | −8.5 [kV] | −6.0 [kV] | 10,000 [ea] |

Table 7 shows the dust collection efficiency results according to the number of brushes. Particle tracing showed a tendency similar to that of the graph of the charging number.

**Table 7.** Collection efficiency according to the number of brushes.

| | Case 1 Brush 2ea | Case 2 Brush 3ea | Case 3 Brush 3ea |
|---|---|---|---|
| Number of particles [ea] | | 10,000 | |
| Filtered count [ea] | 8316 | 8481 | 8818 |
| Number of exits [ea] | 1684 | 1519 | 1182 |
| Efficiency [%] | 83.2 | 84.8 | 88.2 |

These results indicate that the number of carbon brushes has no significant impact on the increase in efficiency. When the distribution of the charging number for particles (Figure 14) was examined in the particle tracing distribution, it was found that approximately 80% of the particles were discharged through the outlet when the charging number was approximately four or less. When it was five or higher, 100% of the particles were collected on the dust collecting plate. The number of particles tended to be similar in each case.

Because there is a limit to the increase in dust collection efficiency when the charging number exceeds a certain value, it is expected that the collection efficiency can be adjusted by the brush location without increasing the number of brushes. Simulation should strive to improve the uncertainty of flow analysis and the quality of interpretation. Compared to the experimental results, the smaller the error rate, the higher the quality of the simulation. Therefore, the experiment was conducted according to the specified test standard.

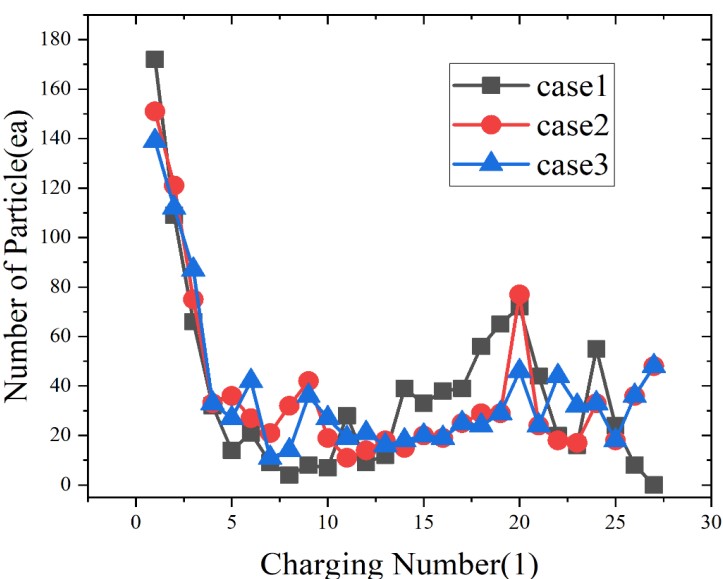

**Figure 14.** Number of particles according to the charging number.

### 4.3. Dust Collection Efficiency Test

To confirm the reliability of the dust collection efficiency simulation results, a test was conducted in a chamber in accordance with the standard of the Korea Air Cleaning Association (SPS-KACA002-0132,2015). The dust collection efficiency evaluation device was fabricated using acrylic products as shown in Figure 15. The device consists of a HEPA filter, an aerosol generator (DS-103, INexus), a test filter, particle counters (Model 11-A, Grimm), and a sirocco fan. In the dust collection test, the collection efficiency of particles with a diameter of 0.3 μm was obtained by introducing a flow velocity of 0.9 m/s and using KCl particles as test particles. The KCl particles were uniformly sprayed using an atomizer in the particle inlet, and the number of particles was measured before and after filtering. The difference was compared to derive the dust collection efficiency.

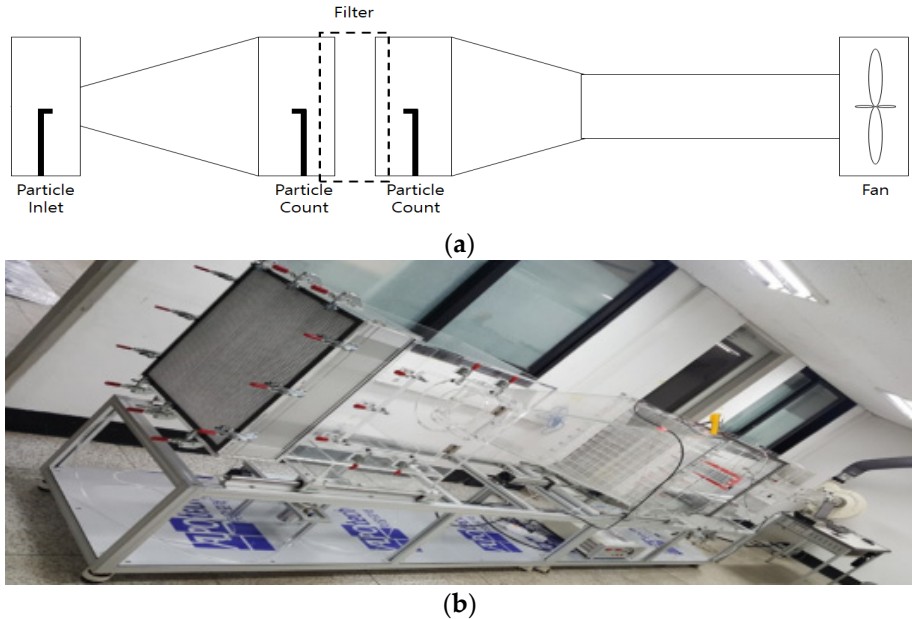

**Figure 15.** Setup of the dust collection efficiency evaluation device; (**a**) schematic diagram; (**b**) wind tunnel dust collection test equipment.

Table 8 and Figure 16 show the results of comparing the simulation and test results. The test results, which were derived by averaging the results of three tests, agreed closely with the simulation results.

**Table 8.** Comparison of efficiency and error rates in the final model.

| | Experiment [%] | | | | Simulation [%] | Error Rate [%] |
|---|---|---|---|---|---|---|
| | **1st** | **2nd** | **3rd** | **Average** | | |
| Case 1 | 80.9 | 82.0 | 81.6 | 81.5 | 83.2 | 2.04 |
| Case 2 | 82.0 | 81.6 | 82.3 | 82.0 | 84.8 | 3.03 |
| Case 3 | 82.1 | 83.6 | 83.3 | 83.0 | 88.2 | 5.89 |

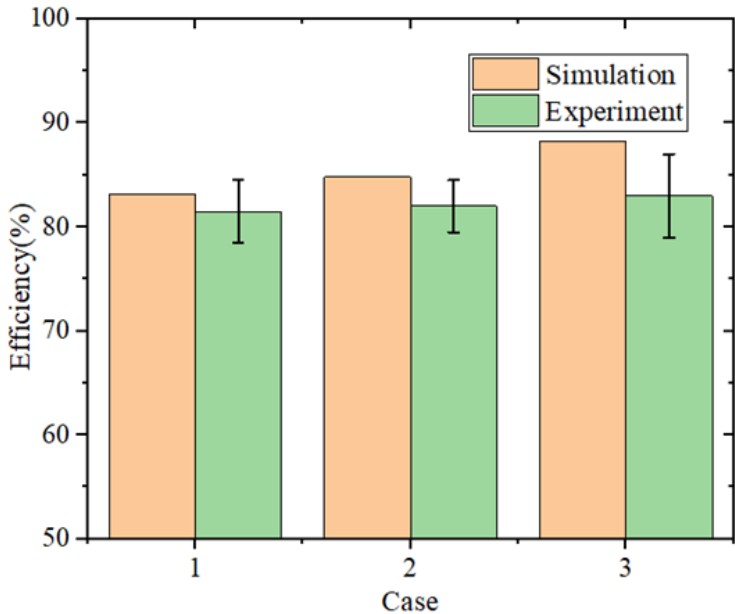

**Figure 16.** Comparison of test results and simulations.

## 5. Conclusions

In this study, an investigation was conducted on an air purifier that applies a high voltage, through experiments and simulations. Carbon brushes were applied to replace the metal applied to existing air purifiers, and a simulation was performed on the ion concentration generated from the carbon brushes.

The reliability of the ion concentration distribution simulation was confirmed by comparing the simulation and experimental results. On the basis of the confirmed reliability, the ion distribution according to the number of brushes was analyzed through simulation. In addition, the error rate was compared by simulating the dust collection efficiency according to the charging number and conducting the filter one-pass test in accordance with SPS-KACA002-0132,2015.

1.  In the ion distribution simulation, a carbon brush to which a negative voltage was applied was simulated, and it was confirmed that the ion distribution diffused in the ground direction. Experiments were performed to examine the reliability of the simulation. An error rate of approximately 4.3% was confirmed, thereby verifying the reliability of the simulation.
2.  When the ion distribution according to the ground position and distance was examined, it was found that ions were generated from the brush and formed in the ground direction. As the distance increased, the ion concentration distribution changed. This finding confirmed that the ground position is an important variable in ion distribution analysis.

3. Dust collection efficiency was simulated on the basis of the charging number calculated according to the brush location, and the efficiency was calculated to be approximately 83.2 to 88.2% depending on the brush configuration. An increase in the number of brushes did not significantly increase the dust collection efficiency, because the particles were discharged through the outlet when the charging number was four or less and 100% of them were collected when it was five or higher. Even when the number of brushes was small, a sufficient dust collection effect could be obtained under the simulation conditions applied in this study.

4. To confirm the reliability of the simulation according to the number of brushes, equipment was designed and fabricated in accordance with the standard of the Korea Air Cleaning Association (SPS-KACA002-0132,2015), and the results were compared with the simulation results. It was found that the dust collection efficiency increased as the number of brushes increased. As with the simulation results, however, the increased rate of dust collection efficiency was not very high as the number of brushes increased. The maximum error rate from the simulation results was found to be 0.836%.

5. It was confirmed that the location of brushes is more important than the number of brushes. The center of the dust collector is judged to be the optimal location of brushes in terms of ion concentration distribution.

Based on this study, it is possible to secure the optimal efficiency for future electrical air purifiers for the removal of ultrafine dust and reduce the burden of actual experiments based on simulations. In the future, it is expected that better results can be obtained by designing and simulating actual products and comparing them.

**Author Contributions:** Conceptualization, H.G.K., S.C.K. and L.K.K.; methodology, L.K.K., S.C.K. and H.G.K.; validation, Y.S.K. and L.K.K.; software, Y.S.K.; formal analysis, S.C.K. and L.K.K.; investigation, Y.S.K.; resources, Y.S.K. and L.K.K.; data curation, Y.S.K.; writing—original draft preparation, Y.S.K.; writing—review and editing, Y.S.K., S.C.K. and L.K.K.; visualization, Y.S.K.; supervision, L.K.K. and H.G.K.; project administration, H.G.K.; funding acquisition L.K.K. and H.G.K. All authors have read and agreed to the published version of the manuscript.

**Funding:** This work was supported by the National Research Foundation of Korea (NRF) funded by the Ministry of Education (NRF-2018R1D1A1B07050752). Also supported by Basic Science Research Program through the National Research Foundation of Korea (NRF) funded by the Ministry of Education (No. 2016R1A6A1A03012069).

**Institutional Review Board Statement:** Not applicable.

**Informed Consent Statement:** Not applicable.

**Data Availability Statement:** Not applicable.

**Conflicts of Interest:** The authors declare no conflict of interest.

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
