# Peer review of "Analysis of Plasma Ion Distribution and Dust Collection Efficiency of Carbon-Brush Air Purifiers"

_applsci, doi:10.3390/app13042101_

Round 1
Reviewer 1 Report
In this paper, a combination of experiments and simulations is used to analyze the plasma ion distribution and dust removal efficiency of air purifier that applies high voltage. Also, the effect of carbon brushes on ion distribution is analyzed and explained. The results show that the brush position is more important than the number of brushes, which provides more accurate design information for the number and position of brushes in air purifiers. However, I cannot recommend to accept this article in current version unless the authors address the following concerns. But before going further, there are still some issues that need to be addressed.
1. The markers of all equations are not exactly right-aligned.
2. The format of references cited in Table 1 is not uniform, for example, [24,25] and [24-25].
3. It is suggested that the plots of ion concentration distributions and charging numbers for the three cases in the article should be put together for comparative analysis, which can be compared more intuitively.
4. In Figure 14, the particle number fluctuates when the Charging Number is about 20. There is no explanation for this phenomenon.
5. In Table 7, the number of brushes for each case should be labeled to show the reader more intuitively.
6. In Figure 16, the horizontal coordinates are incorrectly labeled as Case 2, Case 3, and Case 4.
7. The conclusion section can add some prospect to this research direction if appropriate.
8. Some relative papers may enrich the concepts and background of this work as references: Nat. Commun., 2022, 13: 4867; ACS Appl. Mater. Interfaces, 2021, 13, 52850−52860; Small Methods, 2021, 6, 2101051; Nano Energy, 2019, 63: 103829; Sens. Actuators B: Chem, 2022, 370, 132441.
Author Response
"Please see the attachment."

Reviewer 2 Report
This manuscript investigated the distribution of generated ions in an air purifier with carbon brushes using Computational fluid dynamics (CFD) simulation and lab experiments. The results play down the impact of the number of brushes and highlight the role of the ground position in ion concentration distribution. Although the simulation results are too simple ( based on two-dimensional flow), the study results are pretty informative and useful in offering some context on the design of Carbon-Brush Air Purifiers. Generally, the manuscript was good and easy to follow. I would recommend accepting this paper after addressing the following comments:
- When using CFD simulation, there are quite a lot of user-assumptions involved. Therefore, I recommend presenting the uncertainty and quality of the Computational fluid dynamics simulations in the SI. (Refer to 10.1111/j.1600-0668.2003.00170.x.
- For the lab experiment, why do you neglect the spray electrification phenomenon? The KCl particles are generated using a nebulizer.
- Can you include the protocol for making the KCl particles and their particle size distribution?
- Details about the protocol followed in the atomization process are missing.
- Can the humidity play a role in the experiment? how do you control its effect.
- Line 13: “the measured values” here is confusing. I recommend rewriting the sentence.
- Table 1: most of the values (including Zi ) was obtained from Hind's book; can you clarify how? and add some text in the manuscript about these numbers.
- Line 182: I would recommend changing the abbreviation (dp) here as it is confusing with line 131.
- Line 353: You need to add the details of all used instruments (i.e., name, model, company, and year). The concentration of KCl and the applied pressure as well as the reasoning behind its selection, should be reported here.
- Table 8: the particle number ± standard deviation and the number of samples should be included.
Author Response
"Please see the attachment."

Reviewer 3 Report
The authors have made an analysis of Plasma Ion Distribution and Dust Collection Efficiency of Carbon-Brush Air Purifiers where they have performed simulations of the ion concentration distribution in an air purifier applying high voltage.
The article has a very good structure and good methodological rigour.
I have some comments that may help to improve the quality of the article.
1. The impact articles referenced in the introduction are very poor.
2. Justify why the number of brushes does not affect the dust collection efficiency of approximately 82-83%, so that the location of the brushes turned out to be more important than the number of brushes.
3. What reliability values between numerical and experimental results are achieved by simulating the ion concentration distribution?
4. With an error greater than 4.3% for example; was it possible to reduce the computational cost or the sample size significantly. Why not less than 4.3% error and a larger sample size?
Because the authors are satisfied with an error of 4.3%.
5. How important is the position of the soil in the analysis of the ion distribution in the analysis?
6. Justify in the conclusions why it can be stated that the centre of the dust collector is considered the optimal location of the brushes in terms of ion concentration distribution?
Author Response
"Please see the attachment."

Reviewer 4 Report
The content of the submitted manuscript is good, but the current form's presentation does not fulfill the journal requirements. Modification is needed to consider for publication.
- Title of the paper
The title of the paper looks good, but at the same time, it can be modified to represent the manuscript in a better way.
Abstract
- You should include some of the main findings in the abstract section.
Abstract should have a conclusion of the study.
Introduction
- The objective of the study is also not clearly mentioned.
· Add more on the basics of the problem in the introduction
· More details about identified problems are required in the introduction section.
· The author should focus mainly on the importance and significance of the study.
· I suggest the author demonstrate what the paper adds to the current literature. and what new knowledge is added by this study?
Material and Methods
- The material and method section is too weak in the manuscript, and you need to focus on it more.
Result and discussion
- The presentation fails to discuss the summary and tries to some vague reason which is not an explanation.
Conclusion
- Please rewrite the conclusion with the proper explanation in the R & D.
References
Reference section should be increased with number of recent studies. I would like to suggest to author to include following published article to improve the quality of articles, these are
Is safe distance enough to prevent COVID-19? Dispersion and tracking of aerosols in various artificial ventilation conditions using OpenFOAM. 2022 Gondwana Research (10.1016/j.gr.2022.03.013).
Exposure and health: A progress update by evaluation and scientometric analysis. Stochastic Environmental Research and Risk Assessment 2022, (10.1007/s00477-022-02313-z).
Other comments:
English editing is needed in some parts of the manuscript.
Abbreviations should be explained before the introduction.
Author Response
"Please see the attachment."

Reviewer 5 Report
In this paper, the plasma ion distribution and dust collection efficiency of an air purifier were confirmed according to the location and number of carbon brushes.
As a result of this paper, the ion distribution according to the positions of the five brushes was confirmed, and it was confirmed that there was a difference in ion concentration according to the position. It was confirmed that the ion distribution according to the number of brushes did not have a significant effect. In addition, it was confirmed that the dust collection efficiency was more correlated with the position of the carbon brush than the number.
As a result of reviewing this paper, the experimental and calculation processes were relatively clearly applied. In addition, it is considered that the academic value is sufficient by confirming the error through the dust collection efficiency confirmation. Accordingly, this paper is considered sufficient to be published in this applied science journal after minor revision.
The content of revisions to this paper are as follows.
introduction section
1. Line 50 requires correction of et al. in the citation
2. No citation for Yeo in Line 53
3. There is an introduction to review papers in the introduction section, but the results of the reviewed paper experiments are insufficient, so it needs to be checked in detail and corrected.
Author Response
"Please see the attachment."

Round 2
Reviewer 2 Report
The manuscript has been improved , however, maybe it can be considered more of a scientific issue, and hopefully I’m not overlooking something, but I am still a bit uncomfortable with discounting the uncertainty and quality of the Computational fluid dynamics simulations in this article. At least, this matter should be acknowledged in the manuscript. For the nebulization process, I would strongly recommend referring to https://doi.org/10.1016/j.scitotenv.2019.02.214 and https://doi.org/10.1016/j.scitotenv.2020.140060 as they extensively employed the process to investigate the particulate matter characteristics, including particle number.
Author Response
I acknowledge the uncertainty and quality of the simulation and added a method to discuss the improvement of quality to this paper.
In addition, we added two papers provided to improve the quality of the content.